# Genome Sequence and Characterization of Five Bacteriophages Infecting *Streptomyces coelicolor* and *Streptomyces venezuelae*: Alderaan, Coruscant, Dagobah, Endor1 and Endor2

**DOI:** 10.3390/v12101065

**Published:** 2020-09-23

**Authors:** Aël Hardy, Vikas Sharma, Larissa Kever, Julia Frunzke

**Affiliations:** Institute of Bio- and Geosciences, IBG-1: Biotechnology, Forschungszentrum Jülich, 52425 Jülich, Germany; a.hardy@fz-juelich.de (A.H.); v.sharma@fz-juelich.de (V.S.); l.kever@fz-juelich.de (L.K.)

**Keywords:** phage isolation, phage genomics, *Streptomyces*, *Siphoviridae*, actinobacteriophages, actinorhodin

## Abstract

*Streptomyces* are well-known antibiotic producers, also characterized by a complex morphological differentiation. *Streptomyces*, like all bacteria, are confronted with the constant threat of phage predation, which in turn shapes bacterial evolution. However, despite significant sequencing efforts recently, relatively few phages infecting *Streptomyces* have been characterized compared to other genera. Here, we present the isolation and characterization of five novel *Streptomyces* phages. All five phages belong to the *Siphoviridae* family, based on their morphology as determined by transmission electron microscopy. Genome sequencing and life style predictions suggested that four of them were temperate phages, while one had a lytic lifestyle. Moreover, one of the newly sequenced phages shows very little homology to already described phages, highlighting the still largely untapped viral diversity. Altogether, this study expands the number of characterized phages of *Streptomyces* and sheds light on phage evolution and phage-host dynamics in *Streptomyces*.

## 1. Introduction

*Streptomyces* is a genus of Gram-positive bacteria belonging to the order of Actinobacteria that exhibit a high GC-content (on average about 73 mol% G + C). *Streptomyces* are prolific producers of natural products with a wide range of biological activities. This repertoire of bioactive molecules has been harnessed for medical and agricultural purposes, as for example ⅔ of known antibiotics of microbial origin are produced by *Streptomyces* [1,2,3].

Another distinctive feature of *Streptomyces* is their complex developmental cycle. Unlike most bacteria—that divide by binary fission, *Streptomyces* development is instead centered on the formation of spores. Germinating spores first form a network of interconnected cells, called vegetative mycelium. The vegetative mycelium later serves as a basis for the coordinated erection of an aerial mycelium. This is followed by the segmentation of these aerial filaments into spores, which can then start a new cycle [3,4,5].

Phages infecting *Streptomyces* were described at a quick pace in the 1970–1980s, but most of them were not sequenced later [6,7,8]. The phage phiC31 represents a notable exception to this trend, as it was used to develop crucial genetic tools for *Streptomyces* before being sequenced in 1999 [9,10,11]. Phages R4, SV1, VP5 were also the subject of numerous studies, but the latter was not sequenced [12,13].

*Streptomyces* peculiarities were studied in the context of phage infection. For example, adsorption to mycelium of phage Pal6 was shown to differ depending on the stage of development of *Streptomyces albus* [14]. In this instance, phage adsorption was found to be maximal for germinating spores. Combined with the observation that germinating spores showed an intense average metabolic activity, this suggests that spore germination represents the most sensitive development stage for phage infection.

Conversely, the recent years have seen a sustained effort into the isolation and sequencing of *Streptomyces* phages, notably by the Science Education Alliance-Phage Hunters Advancing Genomics and Evolutionary Science (SEA-PHAGES; https://seaphages.org/) program in the USA [15]. However, few of these phages were extensively characterized.

Here, we report the isolation, characterization and genome analysis of five novel *Streptomyces* phages. Two of them (Alderaan and Coruscant) were isolated using *S. venezuelae*, the remaining three (Dagobah, Endor1 and Endor2) were isolated using *S. coelicolor*. Observation with transmission electron microscopy showed that all five phages belong to the *Siphoviridae* family. Lifestyle prediction with the complete nucleotide sequences revealed that four (Alderaan, Dagobah, Endor1 and Endor2) are probably temperate, while Coruscant was predicted to be a virulent phage. Alderaan, Coruscant, Endor1 and Endor2 show close relatedness to already described *Streptomyces* phages—Endor1 and Endor2 being highly homologous to each other. In contrast, Dagobah showed very little relatedness to any sequenced phage, highlighting the still massively untapped viral diversity.

## 2. Materials and Methods

### 2.1. Bacterial Strains and Growth Conditions

*Streptomyces venezuelae* ATCC 10712 [16] and *Streptomyces coelicolor* M600 [17] and strain M145 [18] were used as main host strains in this study. Cultures were started by inoculating spores from spore stocks stored in 20% glycerol at −20 °C [19]. *S. venezuelae* was grown in liquid Glucose Yeast Malt extract (GYM) medium, while *S. coelicolor* was grown in liquid Yeast Extract Malt Extract (YEME) medium. Unless otherwise stated, cultivation was carried out at 30 °C. For double agar overlays, GYM agar was used for both species, with 0.5% and 1.5% agar for the top and bottom layers, respectively.

### 2.2. Phage Isolation and Propagation

Phages were isolated from soil samples taken near the Forschungszentrum Jülich (Jülich, Germany). Phages contained in soil samples were resuspended by incubation in sodium chloride/magnesium sulfate (SM) buffer (10 mM Tris-HCl pH 7.3, 100 mM NaCl, 10 mM MgSO_4_, 2mM CaCl_2_) for 2 h. The samples were centrifuged at 5000× *g* for 10 min to remove solid impurities. The supernatants were filtered through a 0.22-μm pore-size membrane filter to remove bacteria. For each sample, 1 mL of filtered supernatant was mixed with 3 mL of liquid medium inoculated with 10^7^
*Streptomyces* spores.

After overnight incubation, the culture supernatant was collected by centrifugation at 5000× *g* for 10 min and filtered through a 0.22-μm pore-size membrane filter. Serial dilutions of the filtrate were then spotted on a bacterial lawn propagated by mixing 200 µL of *Streptomyces* overnight culture with 4 mL top agar, according to a modified version of the double agar overlay method [20]. Plaques were visualized after overnight incubation at 30 °C.

Purification of the phage samples was carried out by restreaking single plaques twice [20]. Phage amplification was achieved by mixing 100 µL of the purified phage lysate into top agar to obtain confluent lysis on the plate. After overnight incubation, 5 mL of SM buffer were used to soak the plates and resuspend phages. The resulting phage lysate was centrifuged, and the supernatant was filtered to obtain the high-titer phage solution used for downstream processes.

To assess presence of actinorhodin, the plates were inverted and exposed to ammonia fumes for 15 min by placing 5 mL of 20% ammonium hydroxide solution on the inner surface of the lid.

### 2.3. Electron Microscopy Observation of Phage Virions

For electron microscopy, 5 µL of purified phage suspension were deposited on a glow-discharged formvar carbon-coated nickel grids (200 mesh; Maxtaform; Plano, Wetzlar, Germany) and stained with 0.5% (*wt/vol*) uranyl acetate. After air drying, the sample was observed with a TEM LEO 906 (Carl Zeiss, Oberkochen, Germany) at an acceleration voltage of 60 kV.

### 2.4. Phage Infection Curves

Infection in shake flasks (*S. venezuelae* phages): 70 mL GYM medium were inoculated with 10^5^ spores and incubated at 30 °C for 6–8 h to allow spore germination. Phages were then added at the corresponding multiplicity of infection (MOI). OD_450_ was measured over time to assess bacterial growth. In parallel, the filtered supernatants of the cultures were collected at the same time points. 3 µL of these supernatants were spotted on a *Streptomyces venezuelae* lawn (inoculated to an OD_450_ = 0.4) at the end of the experiment to estimate the phage titer.

Infection in microtiter plates (*S. coelicolor* phages): Growth experiments were performed in the BioLector^®^ microcultivation system of m2p-labs (Aachen, Germany). Cultivation was performed as biological triplicates in 48-well FlowerPlates (m2plabs) at 30 °C and a shaking frequency of 1200 rpm [21]. Backscatter was measured by scattered light with an excitation wavelength of 620 nm (filter module: λ_Ex_/λ_Em_: 620 nm/620 nm, gain: 25) every 15 min. Each well contained 1 mL YEME medium and was inoculated with 10^6^ spores of *S. coelicolor* M145. Phages were added after 7 h, and sampling was performed at the indicated time points. Subsequently, 2 µL of the supernatants were spotted on a lawn of *S. coelicolor* propagated on a double overlay of GYM agar inoculated at an initial OD_450_ = 0.4.

### 2.5. Host Range Determination

The host range of our phages was determined for the following *Streptomyces* species: *S. rimosus* (DSM 40260), *S. scabiei* (DSM 41658), *S. griseus* (DSM 40236), *S. platensis* (DSM 40041), *S. xanthochromogenes* (DSM 40111), *S. mirabilis* (DSM 40553), *S. lividans* TK24 [22], *S. olivaceus* (DSM 41536) and *S. cyaneofuscatus* (DSM 40148). The different *Streptomyces* species were grown in GYM medium, to which glass beads were added to favor dispersed growth.

The host range was determined by spotting serial dilutions of phage solution on lawns of the different *Streptomyces* species, in duplicates. A species was considered sensitive to a given phage only if single plaques could be detected; we further indicated if the phages are able to lyse a species (Table 1).

### 2.6. DNA Isolation

For isolation of phage DNA, 1 µL of 20 mg/mL RNAse A and 1 U/µL DNAse (Invitrogen, Carlsbad, CA, USA) were added to 1 mL of the filtered lysates to limit contamination by host nucleic acids. The suspension was incubated at 37 °C for 30 min. Then, EDTA, proteinase K and SDS were added to the mixture at final concentrations of 50 mM (EDTA and proteinase K) and 1% SDS (*w/v*), respectively. The digestion mixture was incubated for 1 h at 56 °C, before adding 250 µL of phenol:chloroform:isopropanol. The content was thoroughly mixed before centrifugation at 16,000× *g* for 4 min.

The upper phase containing the DNA was carefully transferred to a clean microcentrifuge tube and 2 volumes of 100% ethanol were added as well as sodium acetate to a final concentration of 0.3 M. After centrifugation at 16,000× *g* for 10 min, the supernatant was discarded, and the pellet washed with 1 mL 70% ethanol. Finally, the dried pellet was resuspended in 3 µL DNAse-free water and stored at 4 °C until analyzed.

### 2.7. DNA Sequencing and Genome Assembly

The DNA library was prepared using the NEBNext Ultra II DNA Library Prep Kit for Illumina according to the manufacturer’s instructions and shotgun-sequenced using the Illumina MiSeq platform with a read length of 2 × 150 bp (Illumina). In total, 100,000 reads were subsampled for each phage sample, and de novo assembly was performed with Newbler (GS De novo assembler; 454 Life Sciences, Branford, CT, USA). Finally, contigs were manually curated with Consed version 29.0 [23].

### 2.8. Gene Prediction and Functional Annotation

Open reading frames (ORFs) in the phage genomes were identified with Prodigal v2.6.3 [24] and functionally annotated using an automatic pipeline using Prokka 1.11 [25]⁠. The functional annotation was automatically improved and curated with hidden Markov models (HMMs), and Blastp [26] searches against different databases (Prokaryotic Virus Orthologous Groups (pVOGs) [27], viral proteins and Conserved Domain Database CDD [28]), with the e-value cutoff 10^−10^.

The annotated genomes were deposited in GenBank under the following accession numbers: MT711975 (Alderaan), MT711976 (Coruscant), MT711977 (Dagobah), MT711978 (Endor1) and MT711979 (Endor2). The ends of the phage genomes were determined with PhageTerm [29] using default parameters. Phage lifecycle was predicted with PhageAI [30] using default parameters.

### 2.9. Genome Comparison and Classification

To classify the unknown phage genomes at the nucleotide level, 31 complete reference actinophage genomes belonging to different known clusters were downloaded from the Actinobacteriophage Database [31]. Pairwise average nucleotide identities (ANI) were calculated with the five unknown *Streptomyces* phages and the 31 reference genomes using the python program pyani 0.2.9 [32] with ANIb method. The output average percentage identity matrix file generated from pyani was used for clustering and displayed using the ComplexHeatmap package in R [33]. Phage genome map with functional annotation was displayed using the gggenes package in R.

### 2.10. Protein Domain-Based Classification

An alternative approach was used to classify newly sequenced phages based on conserved protein domains [28]. RPS-BLAST (Reverse PSI-BLAST) searches were performed with e-value cutoff 0.001 against the Conserved Domain Database [28] using the 2486 complete reference actinophages [31], including the newly sequenced phage genomes. Identified Pfam protein domains output files from each phage genome were merged and converted into a numerical presence-absence matrix. The hierarchical clustering dendrogram was constructed with the help of the ward.2 method using the R platform. The resulting dendrogram was visualized using ggtree [34].

## 3. Results

### 3.1. Phage Isolation and Virion Morphology 

Five novel phages infecting *Streptomyces* were isolated from soil samples close to the Forschungszentrum Jülich in Germany. The phages Alderaan and Coruscant were isolated using *Streptomyces venezuelae* ATCC 10712 and formed small, transparent and round plaques of approximately 2 mm of diameter (Figure 1A).

The phages Dagobah, Endor1 and Endor2 were isolated using *Streptomyces coelicolor* M600 as a host strain. Dagobah’s plaques were very small (<1 mm) and were completely formed only after 2 days of incubation. Endor1 and Endor2 formed plaques of 2 mm in diameter with a distinct turbid zone in the center. Additionally, colored halos circling the plaques appeared after 3 days of incubation (Figure 1B). These halos were mostly brownish in the case of Dagobah, and reddish for Endor1 and Endor2. Exposure to ammonia fume resulted in a pronounced blue coloration around plaques, confirming that the halos surrounding plaques contained actinorhodin (Appendix A) [35].

TEM observation of the phage particles revealed that all five phages exhibit an icosahedral capsid and a non-contractile tail (Figure 1C). Based on the morphology, the phages were classified as members of the *Siphoviridae* family.

### 3.2. Infection Curves and Host-Range Determination

Phage infection in liquid cultures was performed to assess infection dynamics. Due to the complex developmental cycle of *Streptomyces*, standard one-step growth curves could not be performed. We instead inoculated liquid cultures with spores of *Streptomyces* and let them germinate for approximately 7 h before adding the phages to a multiplicity-of-infection (MOI) from 0.1 to 10. For the *S. venezuelae* phages, infection was performed in flasks and OD_450_ was used to estimate cell density. In contrast, *S. coelicolor* was cultivated in microtiter plates, and cell growth was monitored using continuous backscatter measurements. In both cases, phage titer was measured over time to estimate the production of phage progeny.

Infection of *S. venezuelae* with Alderaan and Coruscant showed moderate lysis for MOI 1, and distinct OD drops for the MOI 10, which was reduced to almost zero after 24 h of infection (Figure 2A). Phage titers showed a significant increase after 16 h of infection and were markedly higher for MOI 10 than MOI 1.

As for the *S. coelicolor* phages (Figure 2B), infection with Dagobah caused a mild growth delay, visible especially for the highest MOI (MOI 1). In parallel, the phage titers grew moderately (10^2^-fold increase between 0 and 48 h) or strongly (10^5^-fold increase between 0 and 48 h) for initially low (MOI 0.05) or high (MOI 1) MOIs, respectively. In contrast, infection with Endor1 had a profound effect on bacterial growth, as the highest MOIs (MOI 0.1 and 1) effectively suppressed growth. The phage titers showed concordant behavior, with a strong increase from 16 h and a titer plateauing at a high level for MOI 0.1. Endor2 showed an intermediate effect: the growth curves were significantly shifted, proportionally to the initial MOIs. At low MOIs, the evolution of Endor1/2 titer was bell-shaped, with an initial increase until 40 h followed by a decline down to a virtually null titer at the end of the experiment.

Furthermore, the backscatter started to decrease in the uninfected wells starting from 50 h, coinciding with the start of the production of blue-pigmented actinorhodin. A similar drop was also observed in the samples infected with Dagobah, Endor1 and the lowest MOI of Endor2.

Altogether, infection curves revealed that all five phages can successfully propagate in liquid cultures at the expense of their host. Surprisingly, the titers of phages Endor1 and Endor2 dropped after an initial increase, which needs further investigation.

While phages usually have a relatively narrow host range, some phages can sometimes infect many strains of the same species and even distinct species. We assessed the host-range of our phages by spotting them on lawns of different *Streptomyces* species (Table 1).

*S. coelicolor* M145 showed the same sensitivity pattern than the M600 strain. M145 and M600 are both plasmid-free derivatives of A3(2) and mainly differ from each other in the length of their direct terminal repeats [17].

Beside *S. venezuelae* and *S. coelicolor*, *S. lividans* showed plaque formation by phage Dagobah. Endor1 and Endor2 also formed plaques on *S. olivaceus* and *S. cyanofuscatus*. Alderaan, Endor1 and Endor2 caused indefinite clearance of the bacterial lawn of several species, but higher dilutions did not reveal distinct, single plaques. For these species, the phage lysates could have inhibitory effects on growth or cause non-productive infection [36,37].

In summary, Endor1 and Endor2 showed the broadest host range, but overall, the five phages we isolated feature a relatively modest host range, as they are only able to infect few other *Streptomyces* species.

### 3.3. Genome Sequencing and Genome Features

All phages were sequenced using short-read technology (Illumina Mi-Seq). Each genome could be assembled to a single contig, to which >80% of the reads could be mapped confirming the purity of the samples.

The genome features of the five phages are summed up in Table 2. Briefly, they show diverse genome sizes (39 to 133 kb), GC-contents (48 to 72%) and ORFs numbers (51 to 290). The phage Coruscant differed from other phages, in that its genome is significantly larger than the other phages and exhibits a markedly low GC content (48%), in comparison to the one of its host (72%). The genomic ends were predicted using PhageTerm, which detects biases in the number of reads to determine DNA termini and phage packaging mechanisms [29]. Alderaan, Endor1 and Endor2 showed a headful packaging mechanism where the phage genomes have a fixed start at the *pac* site, but the end of the genome is variable. In contrast, phages Coruscant and Dagobah have direct terminal repeats (DTR). These DTR were identified in the initial assembly by an approximately 2-fold increase in coverage clearly delimitated at single base positions. Phage lifecycle was predicted using PhageAI, which developed a lifecycle classifier based on machine learning and natural language processing [30].

Phage genes involved in the same function are usually clustered together, forming functional modules (Figure 3) [38,39]. These modules fulfil the basic functions necessary for production of progeny phages, including DNA/RNA metabolism, DNA replication and repair, DNA packaging, virion structure and assembly (tail and capsid), regulation, lysogeny (in the case of temperate phages) and lysis.

Interestingly, Coruscant’s large genomes is paralleled by a high genome complexity. It contains no less than 41 copies of tRNAs, covering 19 different amino acids—all standard amino acids except valine. Coruscant has also a relatively high fraction of coding sequences for which no function could be predicted (155 hypothetical proteins out 290 CDS compared to 16/51 for Alderaan).

The phages were also found to encode homologs of bacterial regulators that are typically used by *Streptomyces* to control sporulation and overall development. For example, *whiB* (found in Alderaan, and Coruscant) and *ssgA* (found in Dagobah) are both essential for sporulation of *Streptomyces* [40,41]. Three phages (Coruscant, Endor1 and Endor2) also encode Lsr2-like proteins, which are nucleoid-associated proteins functioning as xenogeneic silencing proteins and are conserved throughout Actinobacteria [42].

Additionally, despite overall high synteny and homology, the phages Endor1 and Endor2 showed sequence variations in the tail fiber proteins, tapemeasure and endolysin. In particular, the region encoding distal elements of the tail (ORF_00022 to ORF_00025 in Endor1, ORF_00023 to ORF_00026 in Endor2) displays reduced similarity at the nucleotide level (Appendix A). The resulting differences at the protein level could potentially account for the differences in host range between these two phages, e.g., infectivity on *S. olivaceus* (Table 1).

### 3.4. Average Nucleotide Identity (ANI) Analysis

We established the sequence relationship between the newly sequenced *Streptomyces* phages and the selected genomes from the representative group members of actinophages.

The Average nucleotide identity (ANI) based clustering dendrogram analysis showed that four (Endor1, Endor2, Alderaan, and Coruscant) out of five phage genomes clustered confidently with the members of already known clusters (Endor1/Endor2: BD, Coruscant: BE, and Alderaan: BC) (Figure 4). However, one of the phage genomes (Dagobah) does not share sufficient similarity and was therefore clustered as an unresolved group. Calculation of virus intergenomic similarities using VIRIDIC [43] showed congruent results to the ANI-based clustering (Appendix A), providing further support to the clustering shown in Figure 4. Altogether, the overall analysis showed that except Dagobah, all four phages show close relatedness to Streptomyces phages.

### 3.5. Protein Domain-Based Analysis

Sequence relationship between the phage genomes is most commonly determined with the help of genome-wide similarity or average nucleotide identity-based analysis. However, a traditional method such as phylogeny with single genes is challenging because of the high variability and lack of universal genes across the phage genomes. Thus, we used additional phyletic-based analysis to establish a sequence relationship between the phage genomes. The hierarchical clustering dendrogram based on the identified 703 Pfam domains presence-absence matrix confidently clusters newly sequenced phages with known actinophages (Figure 5).

In comparison to ANI-based analysis, hierarchical clustering showed congruent topology for the four newly sequenced *Streptomyces* phage genomes (Endor1 and Endor2: BD cluster, Alderaan: BC cluster, and Coruscant: BE cluster) (Appendix A). It also resolved polytomy between the unresolved groups and showed that Dagobah comes under the singleton group, consisting of highly divergent phages. Moreover, a high level of congruence was observed between already known groups and the groups identified by our hierarchical clustering. Thus, our results strongly suggest domain-based phyletic or hierarchical clustering analysis as an alternate way of classification of newly sequenced phage genomes.

## 4. Discussion

In this study, we report the isolation and characterization of five novel *Streptomyces* phages. Alderaan and Coruscant were isolated using *S. venezueale*, while *S. coelicolor* was the host used for isolation of Dagobah, Endor1 and Endor2.

The machine-learning based lifestyle prediction tool PhageAI suggested a temperate lifestyle for four of the phages (Alderaan, Dagobah, Endor1 and Endor2) and a virulent lifestyle for Coruscant. These results were congruent with the lifestyle indicated by PhagesDB of phages belonging to the same cluster, as shown by the protein domain-based hierarchical clustering (Appendix A). However, unlike the other members of the BC cluster, Alderaan does not seem to have any integrase domain or gene. Together with the clear plaques it forms, this suggests that this phage potentially lost its integrase and therefore adopted a lytic lifestyle. Such events alter only slightly the overall genome landscape, be it at the nucleotide or protein level, and could thereby explain why whole-genome based predictions like PhageAI or protein domain-based clustering still predict Alderaan as temperate. These discongruencies, however, highlight the requirement of further experimental validation.

In contrast to the other four phages, Coruscant exhibits a large genome (superior to 130 kb) with massive direct terminal repeats (12 kb) and a low GC content (48%), in comparison to the 72% of its *Streptomyces* host. Coruscant also encodes 41 copies of tRNA genes, spanning 19 of the 20 standard amino acids. This large tRNA gene repertoire could be used to optimize gene expression in hosts that have differing codon usage patterns or to counteract potential tRNA-based degradation defense systems [44]. Altogether, the combination of a low GC content and a substantial tRNA equipment suggests a recent adaptation of the phage Coruscant to *Streptomyces*.

ANI and hierarchical clustering analysis revealed that Alderaan, Coruscant and Endor1/Endor2 belong to clusters BC, BE and BD defined by PhagesDB [31], respectively. In contrast, Dagobah showed very little homology with described phages, and was thus considered as a singleton. This finding highlights the largely untapped phage diversity, making the isolation of entirely “novel” phages still possible.

*Streptomyces* are characterized by their complex lifestyle and cellular differentiation. Interestingly, the isolated actinophages also encode homologs of SsgA, WhiB and Lsr2 proteins—regulatory proteins typically encoded by their hosts. The *ssgA* gene product was previously shown to be necessary for proper sporulation of *Streptomyces coelicolor* [41]; *whiB* is also essential for sporulation of *Streptomyces* and was already reported to be found in several actinophages [45,46,47]. Interestingly, the WhiB-like protein of mycobacteriophage TM4, WhiB_TM4_, was shown to inhibit the transcription of *Mycobacterium whiB2.* Expression of WhiB_TM4_ in *M. smegmatis* led to hindered septation resembling a WhiB2 knockout phenotype, highlighting how phage can interfere with their host’s development [46].

Lsr2-like proteins are nucleoproteins conserved in Actinobacteria. In *Streptomyces*, they were recently shown to silence cryptic specialized metabolic clusters [48]. The first example of a phage-encoded Lsr2-like protein is the prophage-encoded Lsr2-like protein CgpS in *Corynebacterium glutamicum* [49]. CgpS was shown to maintain the lysogenic state of the prophage on which it resides. Further bioinformatic searches revealed that Lsr2-like proteins are abundant in actinophages, with almost 20% of *Streptomyces* phages encoding such proteins [42]. However, their role in the coordination of the phage life cycle still remains unclear. Altogether, these observations suggest that phages manipulate their host development, by interfering with central processes such as sporulation and antibiotic production.

More generally, the specificities of *Streptomyces*—especially its morphological complexity—impact the phage isolation and characterization process. For example, the mycelial nature of streptomycetes complicates quantitative studies. The notion of MOI loses a lot of its significance once mycelium has formed, as the network structure originating from one spore has greatly increased phage adsorption but would still be counted as one CFU [14,50]. Furthermore, the formation of clumps, although mitigated by the addition of glass beads or increase of osmotic pressure [51], makes accurate monitoring of cell growth (based on optical density or backscatter) difficult.

*S. coelicolor* was established as a model system for the *Streptomyces* genus partly because of its prolific pigment production [52]. Interestingly, we observed colored halos around the plaques formed by the *S. coelicolor* phages. Exposure to ammonia fume confirmed that these colored halos contain actinorhodin. This observation suggests that *Streptomyces* release metabolites in reaction to phage predation, some of which may potentially have anti-phage properties as it was shown recently with anthracyclines in *Streptomyces peucetius* [53].

Understanding the processes governing phage infection has the potential to illuminate the basic physiology of their hosts. Therefore, phages can serve as a basis to study *Streptomyces*’ specific traits—its complex reproduction cycle and abundant production of secondary metabolites—in the context of phage infection.

## Figures and Tables

**Figure 1 viruses-12-01065-f001:**
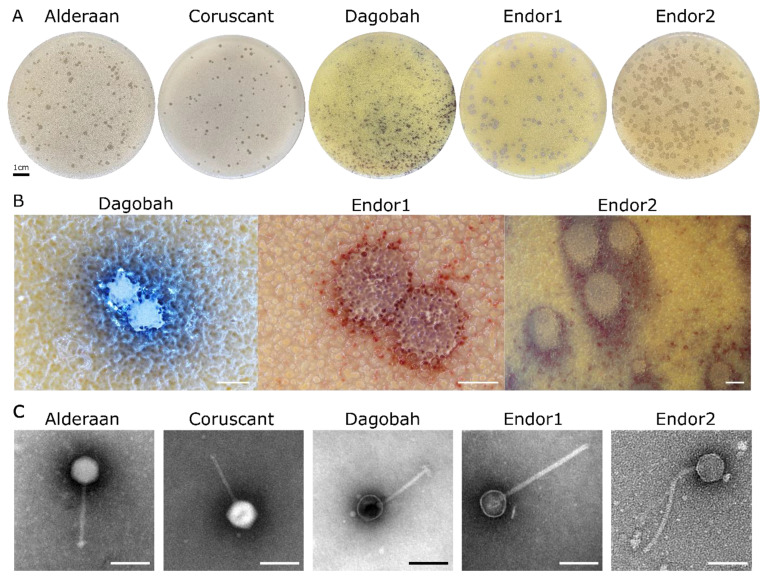
Morphology observation of five novel *Streptomyces* phages. (**A**) Plaque morphologies of the five phages. Double agar overlays were performed to infect *S.venezuelae* ATCC 10712 with the phages Alderaan and Coruscant, and *S. coelicolor* M600 with the phages Dagobah, Endor1 and Endor2. Plates were incubated overnight at 30 °C and another day (3 days in the case of Dagobah) at room temperature to reach full maturity of the bacterial lawn; (**B**) Close-ups of phage plaques imaged using a stereomicroscope Nikon SMZ18. *S. coelicolor* M145 was infected by phages using GYM double agar overlays. The plates were incubated at 30 °C overnight and then kept at room temperature for two (Endor1 and Endor2) or three days (Dagobah). Scale bar: 1 mm; (**C**) Transmission electron microscopy (TEM) of phage isolates. The phage virions were stained with uranyl acetate. Scale bar: 150 nm.

**Figure 2 viruses-12-01065-f002:**
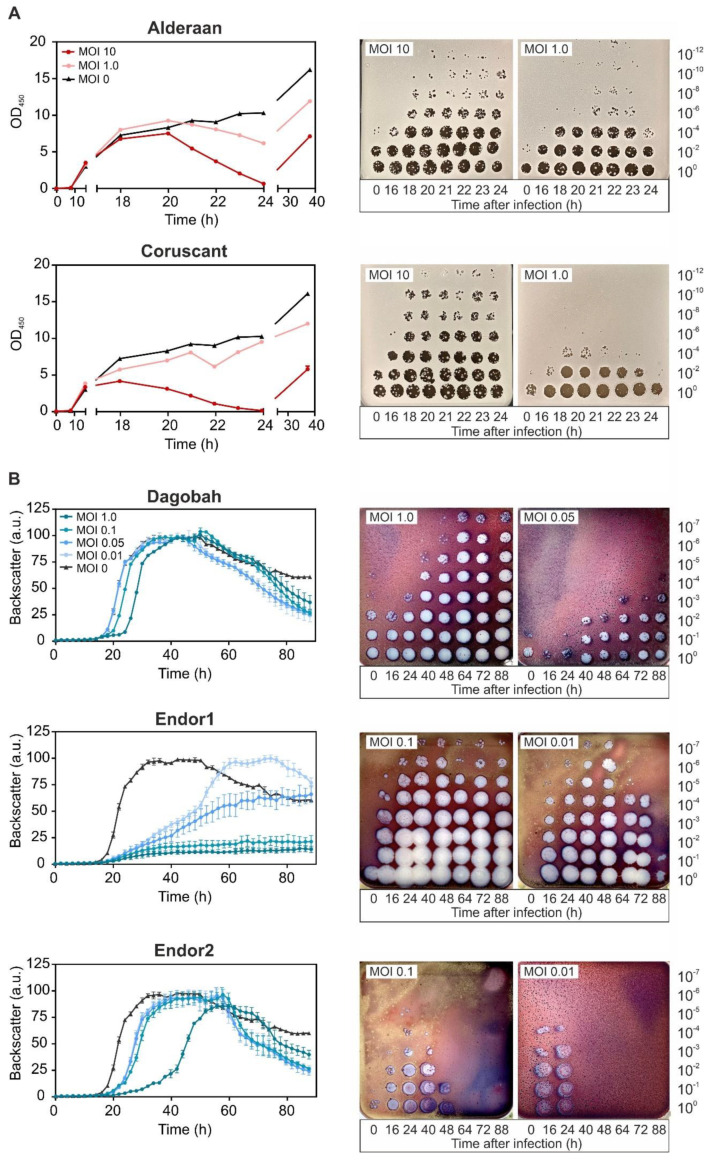
Infection curves of the five phages infecting *S. venezuelae* (**A**) and *S. coelicolor* (**B**)**.** Spores of either *S. venezuelae* (10^5^) or *S. coelicolor* M145 (10^6^) were grown in GYM or YEME medium, respectively. After 6 to 8 h, phages were added at the corresponding multiplicity of infection (MOI). OD_450_ or backscatter were measured over time (left panels), in parallel to phage titers (right panels).

**Figure 3 viruses-12-01065-f003:**
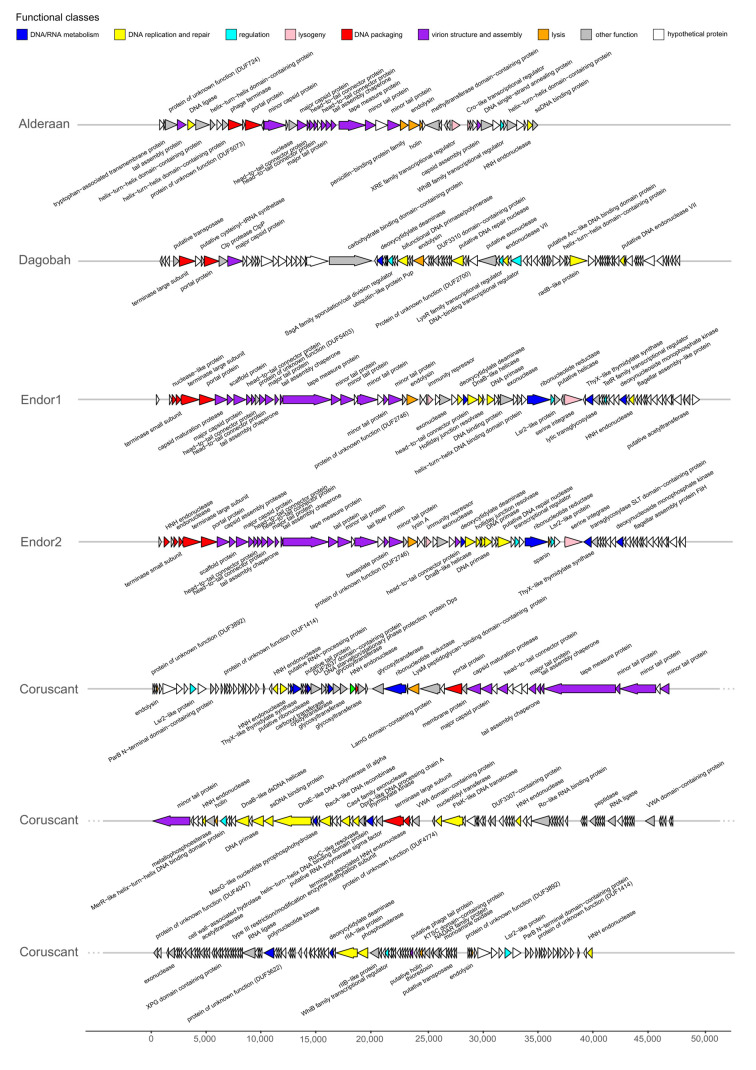
Genome map of the five *Streptomyces* phages. Open reading frames (ORFs) were identified with Prodigal and functionally annotated using an automatic pipeline based on Prokka [25]. The functional annotation was automatically improved and curated using hidden Markov models (HMMs), and Blastp searches [26] against different databases (Prokaryotic Virus Orthologous Groups (pVOGs) [27], viral proteins and Conserved Domain Database (CDD) [28]. Genome maps were created using the R package gggenes.

**Figure 4 viruses-12-01065-f004:**
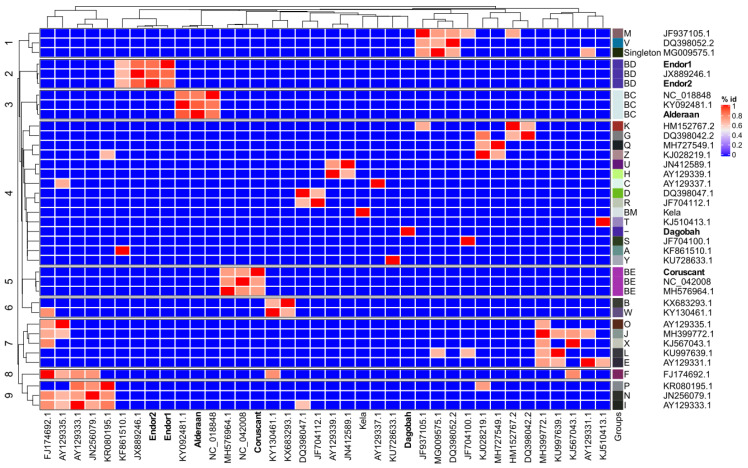
Average nucleotide-based dendrogram analysis using 38 actinophage genomes. These 38 genomes include 31 genomes downloaded from the Actinophage Database (https://phagesdb.org/), two genomes from NCBI based on close relatedness, and the five newly sequenced phages. The group of each phage, as defined by the Actinophage Database, is indicated.

**Figure 5 viruses-12-01065-f005:**
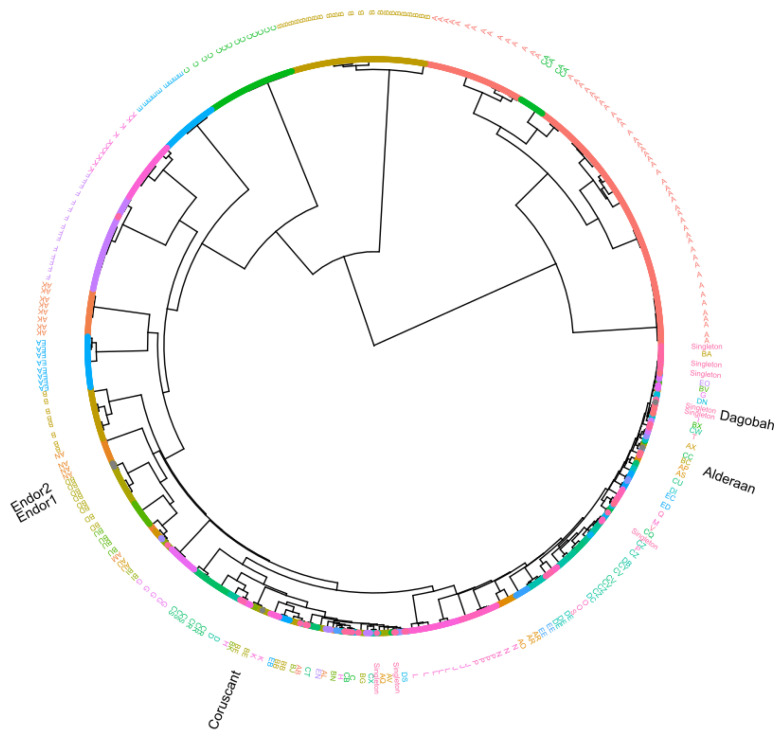
Protein domain-based hierarchical clustering. The dendrogram was constructed based on the presence-absence matrix of the >700 Pfam protein domains identified from 2486 actinophage genomes. Phages are color-coded according to known groups from the Actinobacteriophage Database (https://phagesdb.org/) [31]. The overlapping labels of the outer ring were merged to improve the figure’s readability. The position of the five new phage genomes is indicated as black text.

**Table 1 viruses-12-01065-t001:** The host range of the five phages was assessed by spotting serial dilutions of these phages on lawns of different *Streptomyces* species propagated on GYM medium. The outcome of the spot assays is reported as follows: plaque formation (green), clearance of the bacterial lawn without visible plaques (yellow), no plaque or lysis visible (no color). The efficiency of plating (EOP) of a phage on a given strain relative to the host used for isolation is indicated, when plaques are countable.

	Alderaan	Coruscant	Dagobah	Endor1	Endor2
*S. venezuelae*					
*S. coelicolor M600*					
*S. coelicolor M145*			1	1	1
*S. rimosus subsp. rimosus*					
*S. scabiei*					
*S. griseus*					
*S. platensis*					
*S. xanthochromogenes*					
*S. lividans*			0.2		
*S. olivaceus*					4
*S. cyaneofuscatus*				0.08	0.4

**Table 2 viruses-12-01065-t002:** Basic genome features of the five phages. Open reading frames (ORFs) were predicted using Prokka [25] and were later manually curated. Protein domains encoded in ORFs were identified using RPS-BLAST against the Conserved Domain Database (CDD). The type of genome ends was determined using Phage Term [29]. The lifestyle of each phage was predicted by the machine-learning based program PhageAI [30].

Phage Name	Accession Number	Reference Host	Genome Size (kb)	GC Content (%)	ORF Number	Genome Termini Class	Lifestyle Prediction
Alderaan	MT711975	*Streptomyces venezuelae* ATCC 10712	39	72.1	51	Headful (*pac*)	Temperate
Coruscant	MT711976	*Streptomyces venezuelae* ATCC 10712	133 (12kb DTR)	48.4	290	DTR (long)	Virulent
Dagobah	MT711977	*Streptomyces coelicolor* M600	47 kb (1kb DTR)	68.9	93	DTR (short)	Temperate
Endor1	MT711978	*Streptomyces coelicolor* M600	49	65.8	75	Headful (*pac*)	Temperate
Endor2	MT711979	*Streptomyces coelicolor* M600	48	65.1	75	Headful (*pac*)	Temperate

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
