# Peer review of "Genome Sequence and Characterization of Five Bacteriophages Infecting Streptomyces coelicolor and Streptomyces venezuelae: Alderaan, Coruscant, Dagobah, Endor1 and Endor2"

_viruses, 2020, doi:10.3390/v12101065_

Round 1
Reviewer 1 Report
This paper describes the Genome sequence and characterization of five bacteriophages infecting Streptomyces coelicolor and Streptomyces venezuelae: Alderaan, Coruscant,
Dagobah, Endor1 and Endor2. This manuscript provides new insights into Streptomyces phages.
Overall, this paper is well written, thorough and yet concise. I only have one comment to make.
Item that should be addressed:
Table 1 and corresponding text:
Could the host range susceptibility testing be reported as efficiency of plating (EOP) instead of yes, no, lysis only? An EOP of 1 vs that of not 1 provides valuable information. Reading between the lines of the text, it seems that you have that data.
Also for this table, an undetectable growth could be better represented as EOP instead of no color. It is unclear if no color means no infection noted or not tested.
The difference between Endor1 and Endor2’s ability to infect S. olivaceus is interesting. Can you provide insight into the explanation of that particular difference from the genomic data?
Author Response
Manuscript ID: viruses-918762
Type of manuscript: Article
Title: Genome sequence and characterization of five bacteriophages infecting Streptomyces coelicolor and Streptomyces venezuelae: Alderaan, Coruscant, Dagobah, Endor1 and Endor2
Authors: Ael Hardy, Vikas Sharma, Larissa Kever, Julia Frunzke *
Response to reviewer comments
GE notes:
I would have liked to have seen some experimental confirmation around the lifestyle off the different phages, but this aspect may well be identified during the peer-review process. Figure 3 is of low quality and becomes pixelated when enlarged. The authors might like to consider use of VIRIDIC
rather then ANIb for the analysis of nucleotide sequence identity. The reviewers suggested by the authors are appropriate.
A: The experimental analysis of the phage lifestyles is still work in progress. In Figure S3-6 we added the lifestyle prediction of phages from the same cluster based on the information available from phagesdb.org. We also added a further paragraph on this in the discussion:
“The machine-learning based lifestyle prediction tool PhageAI suggested a temperate lifestyle for four of the phages (Alderaan, Dagobah, Endor1 and Endor2) and a virulent lifestyle for Coruscant. These results were congruent with the lifestyle indicated by PhagesDB of phages belonging to the same cluster, as shown by the protein domain-based hierarchical clustering (Supplementary Figures S3-S6). However, unlike the other members of the BC cluster, Alderaan does not seem to have any integrase domain or gene. Together with the clear plaques it forms, this suggests that this phage potentially lost his integrase and therefore adopted a lytic lifestyle. Such events alter only slightly the overall genome landscape, be it at the nucleotide or protein level, and could thereby explain why whole-genome based predictions like PhageAI or protein domain-based clustering still predict Alderaan as temperate.”
We improved the resolution of Figure 3.
As Figure S2 we added the VIRDIC analysis, which gives very similar results as the ANIb-based analysis.
“Calculation of virus intergenomic similarities using VIRIDIC [43] showed congruent results to the ANI-based clustering (Figure S2), providing further support to the clustering shown in Figure 4.”
Reviewer 1:
This paper describes the Genome sequence and characterization of five bacteriophages infecting Streptomyces coelicolor and Streptomyces venezuelae: Alderaan, Coruscant, Dagobah, Endor1 and Endor2. This manuscript provides new insights into Streptomyces phages.
Overall, this paper is well written, thorough and yet concise. I only have one comment to make.
Item that should be addressed:
Table 1 and corresponding text:
Could the host range susceptibility testing be reported as efficiency of plating (EOP) instead of yes, no, lysis only? An EOP of 1 vs that of not 1 provides valuable information. Reading between the lines of the text, it seems that you have that data.
Also for this table, an undetectable growth could be better represented as EOP instead of no color. It is unclear if no color means no infection noted or not tested.
The difference between Endor1 and Endor2’s ability to infect S. olivaceus is interesting. Can you provide insight into the explanation of that particular difference from the genomic data?
A: We thank the reviewer for these suggestions. We added the EOP values for the strains, which showed single plaques. The absence of color in this table corresponds to the absence of any detectable infection, be it by lysis only or single plaques. We added this information in the caption of the table.
We do not have an explanation to account for the difference of host range between Endor1 and Endor2. Although sharing a high degree of homology, Endor1 and Endor2 are not identical, and difference(s) at the protein level probably explain this disparity in host range.
Reviewer 2 Report
In the manuscript entitled "Genome sequence and characterization of five bacteriophages infecting Streptomyces coelicolor and Streptomyces venezuelae: Alderaan, Coruscant, Dagobah, Endor1 and Endor2" authors describe isolation, morphological and genomic characterization of five novel Streptomyces bacteriophages. Based on comparison of the genome sequences, four phages have homology to already known phages and one has only very little relatedness to any sequenced phage. The data presented expand the knowledge on the interactions between Streptomyces hosts and their phages. The manuscript is well written and the data are clearly presented, however I have several comments and suggestions that could improve it:
- The dimensions of phage heads usually reflect the genome size and should be presented with the TEM photos of phage virions. Although the genome size of the phage Coruscant is about three times bigger than those of other phages, the dimensions of its head in TEM photos seem very similar to others.
- Presenting the host range of the phages in colour code does not seem very good solution, because the grey colour (hosts used for phage isolation) should also be green (plaque formation). Furthermore, it is not clear from the yellow colour at what titer the clearance of the bacterial lawn is visible.
- I was not able to find the genome sequences according to the accession numbers indicated in the text, so I cannot comment on the annotations of phage genomes.
- The “Results” section is somewhat descriptive in style, and lacks interpretations of the results. For example, infection curves presented in the Figure 2 are described in detail in the text, but it is not explained what conclusions can be made from these data.
Author Response
Reviewer 2
In the manuscript entitled "Genome sequence and characterization of five bacteriophages infecting Streptomyces coelicolor and Streptomyces venezuelae: Alderaan, Coruscant, Dagobah, Endor1 and Endor2" authors describe isolation, morphological and genomic characterization of five novel Streptomyces bacteriophages. Based on comparison of the genome sequences, four phages have homology to already known phages and one has only very little relatedness to any sequenced phage. The data presented expand the knowledge on the interactions between Streptomyces hosts and their phages. The manuscript is well written and the data are clearly presented, however I have several comments and suggestions that could improve it:
The dimensions of phage heads usually reflect the genome size and should be presented with the TEM photos of phage virions. Although the genome size of the phage Coruscant is about three times bigger than those of other phages, the dimensions of its head in TEM photos seem very similar to others.
A: While we agree that capsid size generally reflects the size of the phage genetic material, there are several exceptions to this rule. For example, jumbo phages with genomes close to 300kb can have capsid whose diameter does not exceed 130nm.1,2
In the case of the Coruscant phage, phages belonging to the same BE cluster-as defined by PhagesDB-such as phage NootNoot3 show a capsid smaller than 100nm although displaying similar genomes of 130kb.
Furthermore, differences in packaging strategies and/or packaging efficiencies could account for this seemingly paradoxical observation that capsids of similar size can accommodate sometimes very different amounts of genetic material.
- https://www.frontiersin.org/articles/10.3389/fmicb.2018.02501/full
- https://www.frontiersin.org/articles/10.3389/fmicb.2019.02772/full
- https://phagesdb.org/phages/NootNoot/
Presenting the host range of the phages in colour code does not seem very good solution, because the grey colour (hosts used for phage isolation) should also be green (plaque formation). Furthermore, it is not clear from the yellow colour at what titer the clearance of the bacterial lawn is visible.
A: We agree that the color scheme used initially was not ideal, and we therefore used green instead of grey to indicate plaque formation on the host used for isolation, as suggested. Furthermore, we added the EOP values for the strains, which showed single plaques. We added this information in the caption of the table.
Regarding the use of yellow color for bacterial lysis without plaque formation, lysis was obtained in most cases only with the undiluted phage lysate (the titers of the solutions spotted were approximately comprised between 106 and 108 PFU/ml). However, we are not sure that indicating this information would be valuable for the reader, since the absence of plaque formation is generally used as an indicator of a failure to perform productive infection.
I was not able to find the genome sequences according to the accession numbers indicated in the text, so I cannot comment on the annotations of phage genomes.
The genomes were not accessible on NCBI at the time of submission, and we apologize for this. This has been fixed in the meantime, and accession numbers are now functional.
The “Results” section is somewhat descriptive in style, and lacks interpretations of the results. For example, infection curves presented in the Figure 2 are described in detail in the text, but it is not explained what conclusions can be made from these data.
A: As suggested, we added the following conclusion for the infection curves:
“Altogether, infection curves revealed that all 5 phages can successfully propagate in liquid cultures at the expense of their host. Surprisingly, the titers of phages Endor1 and Endor2 dropped after an initial increase, which needs further investigation.”